# Don't be so negative! Score-based Generative Modeling with Oracle-assisted Guidance

## Abstract

The maximum likelihood principle advocates parameter estimation via optimization of the data likelihood function. Models estimated in this way can exhibit a variety of generalization characteristics dictated by engineering choices such as architecture, parameterization, and optimization bias. This work addresses model learning in a setting where, in addition to the training dataset, there further exists side-information in the form of an oracle that can label samples as being outside the support of the true data generating distribution. Specifically we develop a new denoising diffusion probabilistic modeling methodology, Gen-neG, that leverages this additional side-information. Gen-neG builds on classifier guidance in diffusion models to guide the generation process towards the positive support region indicated by the oracle. We empirically establish the utility of Gen-neG in applications including collision avoidance in self-driving simulators and safety-guarded human motion generation.

## 1 Introduction

What should we do when we train a generative model that produces samples that we know are "bad" or "not allowed?" For instance, when generating traffic scenes, the condition of road users overlapping one another is considered unacceptable. Likewise, in robotics, numerous physics-based constraints must be upheld in the motion and configuration of the robot. Typically, generative models are trained only on a set of "good" training data samples by maximizing their likelihood under the learned model. Nevertheless, when sampling from a fully trained, highly expressive model, some fraction of generated samples fall into the category of "bad" samples. Here we consider the problem of generative modeling when, in addition to the typical training dataset of "good" samples, we are also given access to constraints in the form of an oracle that asserts whether a sample is "bad." Such oracles are ubiquitous in practice and are often a simple function implemented by a domain expert.

The most natural way to use an oracle is to deploy the model in a rejection sampling loop in which the oracle is used to decide whether to reject a sample or not. Depending on circumstances this may be an acceptable final "generative model", but it may come at an unacceptable computational cost. A more effective approach to the problem is to directly minimize the rate of bad samples generated, by restricting the generative model to only place mass on the positive support region allowed by the oracle.

Modern highly expressive deep generative models are sufficiently parameterized and easy to optimize so that they can effectively be trained to result in the de facto non-parametric optimal maximum likelihood solution of placing a mixture of Dirac measures directly on the training data. Transformer-based (Vaswani et al., 2017) denoising diffusion process models (Sohl-Dickstein et al., 2015; Ho et al., 2020) are one-such model class. Such models must be either trained on tremendously large amounts of data (Rombach et al., 2022) or otherwise "de-tuned" (smaller architecture, fewer integration steps, etc.) to ensure that they generalize rather than memorize (Zhang et al., 2021; Arpit et al., 2017). In this paper we are agnostic to this point and assume that we are operating in a realistic modeling regime where the model generalizes. Our work can be seen as a way to control the specific kind of generalization that the model exhibits.

We focus specifically on enabling learning with constraints in score-based models. What we discovered, and report in this work, is a *simple rule for classifier guidance* that has been seemingly, to the best of our knowledge, surprisingly overlooked. Drawing inspiration from results in the very

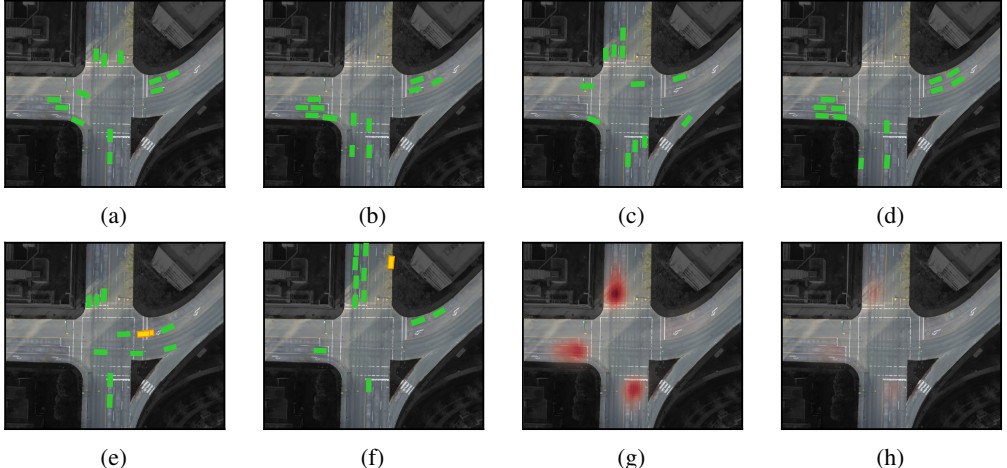

Figure 1: Gen-neG applied to a diffusion model of non-infracting static vehicle placements (i.e. diffusing in the space of a set of oriented rectangles) for the efficient initialization of autonomous vehicle planning simulators (see Zwartsenberg et al. (2023) for a similar model and full problem description). The top row show samples (green "cars") that are not colliding (non-overlapping) and not off-road (stay within the unshaded area of road surfaces) from a baseline diffusion model improved by Gen-neG. The second row shows the kind of infractions our oracle identifies as not being in the support of the true distribution; (e): a collision (yellow overlapping "cars"), (f): (yellow off-road "car"). (g) and (h) graphically illustrate the reduction in infractions per unit area before and after Gen-neG is applied to the baseline model (both plots are normalized to the same maximum value). Quantitative results corresponding to this plot appear later in Table 1.

recent work on discriminator guidance in diffusion processes (Kim et al., 2022), we establish a new methodology for classifier guidance that learns and uses a sequence of differentiable classifiers fit to synthetic samples labelled by the oracle drawn from a sequence of classifier guided diffusion models. The resulting sequence of multiply classifier guided diffusion models (or the end of a sequence of distilled models; details to follow) monotonically decreases the rejection rate while empirically maintaining a competitive probability mass assigned to validation samples. The key insight and technical contribution boils down to carefully choosing the ratio of "good" and "bad" synthetic samples to use when training each of the sequence of guidance classifiers.

We demonstrate our methodology, which we call **Gen**erative modeling with **neG**ative examples (Gen-neG) on several problems, including modeling motion capture sequence data in a way that eliminates ground plane violations and static traffic scene vehicle arrangements that avoid collisions and off-road placements.

## 2 BACKGROUND

### 2.1 SCORE-BASED DIFFUSION MODELS

Score-based diffusion models (Sohl-Dickstein et al., 2015; Song & Ermon, 2019; Ho et al., 2020; Song et al., 2021), also referred to as diffusion models (DMs) are a class of generative models that are defined through a stochastic process which gradually adds noise to samples from a data distribution $q_0(\mathbf{x}_0)$, such that when simulated forward from $t = 0$ the marginal distribution at time $T$ is $q_T(\mathbf{x}_T) \approx \pi(\mathbf{x}_T)$ for some known $\pi(\mathbf{x}_T)$ typically equal to $\mathcal{N}(\mathbf{0}, \boldsymbol{I})$. This is known as the "forward process" and is formulated as an SDE

$$d\mathbf{x}_t = f(\mathbf{x}_t, t)dt + g(t)d\mathbf{w}, \quad \mathbf{x}_0 \sim q_0(\mathbf{x}_0), \tag{1}$$

where $f$ and $g$ are predefined drift and diffusion coefficients of $\mathbf{x}_t$ and $\mathbf{w}$ is the standard Wiener process. DMs generate data by learning the inverse of this process, which is known as the "reverse

process" and defined as

$$dx_t = [f(x_t, t) - g(t)^2 s_\theta(x_t; t)]d\bar{t} + g(t)d\bar{w}, \quad x_T \sim \pi(x_T), \tag{2}$$

where $\bar{t}$ and $\bar{w}$ are the infinitesimal reverse time and reverse Wiener process, respectively. If $s_\theta$ is equal to the score function of the marginals of the forward process, the terminal distribution of the reverse process coincides with $q_0(x_0)$ (Anderson, 1982). Formally,

$$s_\theta(x_t; t) = \nabla_{x_t} \log q_t(x_t) \Rightarrow p_\theta(x_0; 0) = q_0(x_0), \tag{3}$$

where $p_\theta(x_t; t)$ is the marginal distribution of the approximate reverse process.

In order to approximate the score function $\nabla_{x_t} \log q_t(x_t)$, DMs minimize the following score matching objective function (Hyvärinen & Dayan, 2005; Vincent, 2011; Song & Ermon, 2019):

$$\mathcal{L}_\theta^{\text{DM}} = \mathbb{E}_{t, x_0, x_t} \left[ \gamma_t \| s_\theta(x_t; t) - \nabla_{x_t} \log q(x_t|x_0) \|^2 \right], \tag{4}$$

where $x_0 \sim q(x_0)$, $x_t \sim q(x_t|x_0)$, $t$ is sampled from a distribution over $[0, T]$, and $\gamma_t$ is a positive weighting term. Importantly, the Wiener process in Eq. (1) allows direct sampling from the marginals of the forward distributions (Song et al., 2021), i.e. $q(x_t|x_0) = \mathcal{N}(\alpha_t x_0, \sigma_t)$, with $\alpha_t$ and $\sigma_t$ determined by the drift and diffusion coefficients in Eq. (1). This formulation moreover allows the evaluation of the conditional score function ($\nabla_{x_t} \log q(x_t|x_0)$) in closed form.

Many of the DMs reported in the literature operate on discrete time steps (Ho et al., 2020; Song et al., 2020; Nichol & Dhariwal, 2021), and can be considered as particular discretizations of the presented framework. Various parameterizations of the score function have been also explored in the literature (Karras et al., 2022).

In the remainder of this paper we use $q$ to denote the forward process, $s_\theta$ for the score function of the reverse process and $p_\theta$ as the distribution generated by running Eq. (2) backward in time. This applies to the marginals, conditionals, and posteriors as well. Furthermore, to avoid unnecessary notation clutter throughout the rest of the paper, we omit the explicit mention of $\theta$ and $\phi$ and $t$ when their meaning is evident from the context.

## 2.2 CLASSIFIER GUIDANCE

A distinctive and remarkable property of DMs is the ability to train an unconditional version and sample from its class-conditional distributions at inference time without requiring re-training or fine-tuning (Dhariwal & Nichol, 2021; Song et al., 2021). However, it requires a time-dependent classifier $q(y|x_t) = \int q(y|x_0)q(x_0|x_t)\,dx_0$. Here, $q(y|x_0)$ is a traditional classifier, that predicts the class probabilities for each $y$ given a datum $x_0$ from the dataset. While $q(y|x_t)$ classifies a noisy datum $x_t$ sampled from $q_t(x_t) = \int q(x_t|x_0)q(x_0)\,dx_0$.

Classifier guidance follows from the identity $\nabla_{x_t} \log q(x_t|y) = \nabla_{x_t} \log q(x_t) + \nabla_{x_t} \log q(y|x_t)$. If $s_\theta(x_t; t)$ is the score function of the DM, then the score function of the class-conditional DM is

$$s_\theta(x_t|y; t) = s_\theta(x_t; t) + \nabla_{x_t} \log q(y|x_t). \tag{5}$$

**Binary classification** A special case of the above classifier guidance that we use in this paper is when there are only two classes. We provide here a brief overview of such a binary classification task and the notation associated with it. Let $q(x|y = 1)$ and $q(x|y = 0)$ be the distribution of positive and negative examples. Let $\alpha$ and $1 - \alpha$ be the prior probabilities $q(y)$ of positive and negative examples. We then have $q(x) = q(y = 1)q(x|y = 1) + q(y = 0)q(x_t|y = 0) = \alpha q(x_t|y = 1) + (1 - \alpha)q(x_t|y = 0)$. A binary classifier $C_\phi : \mathcal{X}, [0, T] \to [0, 1]$, can then be trained to approximate $q(y = 1|x_t)$ by minimizing the expected cross-entropy loss

$$\mathcal{L}_\phi^{\text{CE}} = -\mathbb{E}_t \left[ \mathbb{E}_{q(x_t)} \left[ q(y = 1|x_t) \log C_\phi(x_t; t) + q(y = 0|x_t) \log(1 - C_\phi(x_t; t)) \right] \right]. \tag{6}$$

Minimizing the cross-entropy loss between the classifier output and the true label is equivalent to minimizing the KL divergence between the classifier output and the Bayes optimal classifier (see Appendix A.3). The minimizer of this loss is then

$$C_{\phi^*}(x_t; t) = \frac{\alpha q(x_t|y = 1)}{\alpha q(x_t|y = 1) + (1 - \alpha)q(x_t|y = 0)}. \tag{7}$$

## 3 METHODOLOGY

In this section, we describe **Gen**erative modelling with **neG**ative examples (Gen-neG). Gen-neG makes use of an oracle that can distinguish samples that are outside of the support for the problem. In a nutshell, Gen-neG consists of first training a DM on given training data as usual, (we refer to the resulting DM as the "baseline DM" throughout). Then we draw samples from this baseline DM, label them using the oracle, then train a classifier (for guidance) using those samples, critically, as we will demonstrate, ensuring that the correct ratio of positive and negative examples are used in training. We then combine this classifier and the baseline model in the typical classifier guidance way. Gen-neG establishes this classifier guided DM as a new baseline DM (either fixing and ultimately "stacking" classifiers or optionally distilling the classifier guided DM into a DM with a single score function estimator) and then repeats this process of generating samples from this new baseline, labelling them with the oracle, training a classifier with the carefully chosen label ratio, employing classifier guidance, stacking or distilling, then repeating. Throughout we will refer to the resulting DM as stacked or distilled depending on whether or not distillation into a unified score function is employed.

### 3.1 PROBLEM FORMULATION AND NOTATION

Let $\mathcal{D} = \{\mathbf{x}^i\}_{i=1}^N \sim q(\mathbf{x})$ be a dataset of i.i.d. samples from an unknown data distribution $q$. Furthermore, let $\mathcal{O} : \mathcal{X} \to \{0, 1\}$ be an oracle function that assigns each point in the data space $\mathcal{X}$ a binary label. In other words, this oracle partitions the data space into two disjoint sets $\mathcal{X} = \Omega \cup \Omega^{\complement}$ such that $\mathcal{O}(\mathbf{x}) = \mathbf{1}_{\Omega}(\mathbf{x})$. Our objective is to learn a score-based diffusion model that (1) maximizes the likelihood of $\mathcal{D}$ and (2) avoids allocating probability to $\Omega^{\complement}$.

In the first stage of Gen-neG we train a DM, $p_\theta(\mathbf{x})$, following standard DM training procedures (e.g. Section 2.1) without utilizing the oracle. In the second stage, we leverage the oracle to train a binary classifier that guides the generation process of DM to avoid $\Omega^{\complement}$. We explain this second stage in the rest of this section.

### 3.2 BAYES OPTIMAL CLASSIFIER GUIDANCE FOR DIFFUSION MODELS

The core component of Gen-neG is a classifier that discriminates between positive and negative samples respectively in $\Omega$ and $\Omega^{\complement}$, which is used to guide the baseline DM. There are two main insights required for Gen-neG that our work provides, the first of which is how to obtain the correct distribution of training data for such a classifier and the second is how to train a classifier which does not shift the sampling distribution.

**Oracle-assisted classifier guidance** Classifier guidance in score-based models is typically used to generate samples from a specific pre-defined class on the training dataset. For instance, a classifier trained on an image classification dataset can be utilized to guide an unconditional diffusion model that has been trained on the same dataset. Unlike traditional approaches that rely on explicit, predefined, labelled datasets, our framework operates based on the oracle function $\mathcal{O}(\mathbf{x})$, which determines the validity of samples.

Gen-neG instead builds a binary classification task using the fully-synthetic data generated by the baseline DM i.e. the data is distributed as $p_\theta(\mathbf{x})$ and the labels are $y = \mathcal{O}(\mathbf{x})$. A time-dependent binary classifier $C_\phi$ is then trained on this dataset. Finally the classifier is incorporated into the baseline DM by

$$s_{\theta,\phi}(\mathbf{x}_t; t) = s_\theta(\mathbf{x}_t; t) + \nabla_{\mathbf{x}_t} \log C_\phi(\mathbf{x}_t; t). \tag{8}$$

Equivalently, we denote the marginal distributions generated by the oracle-assisted DM as $\tilde{p}_{\theta,\phi}(\mathbf{x}_t; t)$. In the rest of this section we show why the classifier guidance in $\tilde{s}_{\theta,\phi}$ helps to enhance the model and reduce the amount of mass on $\Omega^{\complement}$, i.e. $\int_{\mathbf{x} \in \Omega^{\complement}} p_{\theta,\phi^*}(\mathbf{x})$.

**Theorem 1.** *Let $p_\theta(\mathbf{x})$ be the distribution learned by a baseline DM with marginal distributions denoted by $p_\theta(\mathbf{x}_t; t)$ and let $p_\theta(y = 1|\mathbf{x}_0) = \mathcal{O}(\mathbf{x}_0)$. Further, let $C_{\phi^*} : \mathcal{X}, [0, T] \to [0, 1]$ be the Bayes-optimal time-dependent binary classifier arising from perfectly optimizing the following cross-entropy objective*

$$\mathcal{L}_\phi^{CE} = -\mathbb{E}_t \left[ \mathbb{E}_{p_\theta(\mathbf{x}_0, \mathbf{x}_t)} \left[ \mathcal{O}(\mathbf{x}_0) \log C_\phi(\mathbf{x}_t; t) + (1 - \mathcal{O}(\mathbf{x}_0)) \log(1 - C_\phi(\mathbf{x}_t; t)) \right] \right] \tag{9}$$

**Algorithm 1** Gen-neG

1: Input: dataset $\mathcal{D}$, oracle $\mathcal{O}$, balanced synthetic dataset size $N$
2: $i \leftarrow 0$
3: $\theta_i \leftarrow \arg\min_\theta \mathcal{L}_\theta^{\text{DM}}$ $\qquad\qquad\qquad\qquad\qquad\qquad$ ▷ train baseline DM, Eq. (4)
4: $s_i \leftarrow s_{\theta_i}(\mathbf{x}_t; t)$
5: **while** not done **do**
6: $\quad$ $\mathcal{D}_i^+, \mathcal{D}_i^- \leftarrow$ generate samples from DM with score function $s_i$ and label with $\mathcal{O}$
7: $\quad$ **while** $\min(|\mathcal{D}_i^+|, |\mathcal{D}_i^-|) < N$ **do**
8: $\quad\quad$ $\mathcal{D}^+, \mathcal{D}^- \leftarrow$ generate more samples from DM with score function $s_i$ and label with $\mathcal{O}$
9: $\quad\quad$ $\mathcal{D}_i^+ \leftarrow \mathcal{D}_i^+ \cup \mathcal{D}^+, \mathcal{D}_i^- \leftarrow \mathcal{D}_i^- \cup \mathcal{D}^-$
10: $\quad$ **end while**
11: $\quad$ $\alpha_i \leftarrow |\mathcal{D}_i^+|/(|\mathcal{D}_i^+| + |\mathcal{D}_i^-|)$ ▷ Estimate class prior probabilities for Bayes optimal classifier
12: $\quad$ $\mathcal{D}_i^+ \leftarrow \text{select}(N, \mathcal{D}_i^+), \mathcal{D}_i^- \leftarrow \text{select}(N, \mathcal{D}_i^-)$ $\qquad$ ▷ balance dataset for IS classifier training
13: $\quad$ $\phi_i \leftarrow \arg\min_\phi \hat{\mathcal{L}}_\phi^{\text{cls}}(\alpha_i, \mathcal{D}_i^+, \mathcal{D}_i^-)$ $\qquad\qquad\qquad$ ▷ train guidance classifier, Eq. (12)
14: $\quad$ $i \leftarrow i + 1$
15: $\quad$ **if** distill **then**
16: $\quad\quad$ $\psi \leftarrow \arg\min_\psi \mathcal{L}_\psi^{\text{dtl}}$ $\qquad\qquad\qquad\qquad\qquad$ ▷ distill into single DM, Eq. (13)
17: $\quad\quad$ $s_i \leftarrow s_\psi(\mathbf{x}_t; t)$
18: $\quad$ **else** $\qquad\qquad\qquad\qquad\qquad\qquad\qquad\qquad\qquad$ ▷ "stack" guidance classifiers
19: $\quad\quad$ $s_i \leftarrow s_{i-1} + \nabla_{\mathbf{x}_t} \log C_{\phi_i}(\mathbf{x}_t; t)$ $\qquad\qquad\qquad\qquad$ ▷ See Eq. (8)
20: $\quad$ **end if**
21: **end while**
22: **return** DM score function $s_i$

---

*then*

$$\nabla_{\mathbf{x}_t} \log p_\theta(\mathbf{x}_t; t) + \nabla_{\mathbf{x}_t} \log C_{\phi^*}(\mathbf{x}_t; t) = \nabla_{\mathbf{x}_t} \log p_\theta(\mathbf{x}_t | y = 1; t). \tag{10}$$

In other words, by using a Bayes-optimal binary classifier for guidance, we target exactly the score function of positive (oracle-approved) examples.

**Corollary 1.1.** *For an optimal classifier $C_{\phi^*}$,*

1. *$p_{\theta, \phi^*}(\mathbf{x}) = p_\theta(\mathbf{x}|y = 1)$,*

2. *There is no mass on $\Omega^{\complement}$, i.e. $\int_{\mathbf{x} \in \Omega^{\complement}} p_{\theta, \phi^*}(\mathbf{x}) = 0$,*

3. *For any dataset $\mathcal{D} \subseteq \Omega$, $p_{\theta, \phi^*}(\mathcal{D}) \geq p_\theta(\mathcal{D})$.*

Corollary 1.1 suggests the guidance our Gen-neG methodology can improve the baseline DM in terms of both infraction rate and test dataset likelihood.

See the proofs for the Theorem 1 and Corollary 1.1 in Appendices A.1 and A.2.

**Training the classifier** Training the classifier in our approach presents a noteworthy challenge due to the major label imbalance within the synthetic dataset $\mathcal{D}$ generated by the model. This imbalance emerges because the baseline is already close to the target distribution, resulting in a scarcity of negative examples. However, these negative examples play a crucial role in guiding the model at the boundary between positive and negative examples, where the model requires the most guidance.

Gen-neG addresses this challenge by sampling a balanced dataset $\mathcal{D}$ from the model, ensuring the same number of positive and negative examples. However, Gen-neG crucially employs importance sampling in the classifier's training objective to rectify the bias introduced by having to balance the dataset to achieve high classifier accuracy in training. Theorem 1 suggests that a classifier trained on dataset whose marginal distribution over labels differs from the true marginal distribution over labels will target the wrong cross-entropy and arrive at a classifier guided DM that does not necessarily target the distribution of interest. We show evidence of this happening in Fig. 2.

Finally, in order to avoid the computational cost of sampling from the baseline DM to compute the classifier objective in Eq. (9), we approximate the $p(\mathbf{x}_0, \mathbf{x}_t) = p(\mathbf{x}_0)p(\mathbf{x}_t|\mathbf{x}_0) \approx p(\mathbf{x}_0)q(\mathbf{x}_t|\mathbf{x}_0)$.

This is similar to the approximation in Kim et al. (2022). In particular, the classifier's objective in Gen-neG is

$$\mathcal{L}_\phi^{\text{cls}}(\alpha) := \alpha \mathbb{E}_{p_\theta(\mathbf{x}_0|y=1)} \left[ \mathbb{E}_{q(\mathbf{x}_t|\mathbf{x}_0)} \left[ -\log C_\phi(\mathbf{x}_t; t) \right] \right]$$
$$+ (1-\alpha) \mathbb{E}_{p_\theta(\mathbf{x}_0|y=0)} \left[ \mathbb{E}_{q(\mathbf{x}_t|\mathbf{x}_0)} \left[ -\log(1 - C_\phi(\mathbf{x}_t; t)) \right] \right]. \tag{11}$$

Given the balanced dataset $\mathcal{D} = \mathcal{D}^+ \cup \mathcal{D}^-$ where $\mathcal{D}^+ \sim p(\mathbf{x}_0|y=1)$, $\mathcal{D}^- \sim p(\mathbf{x}_0|y=0)$, and $N = |\mathcal{D}^+| = |\mathcal{D}^-|$,

$$\hat{\mathcal{L}}_\phi^{\text{cls}}(\alpha, \mathcal{D}^+, \mathcal{D}^-) := \frac{1}{N} \sum_{\mathbf{x}_0 \in \mathcal{D}^+} \alpha \mathbb{E}_{q(\mathbf{x}_t|\mathbf{x}_0)} \left[ -\log C_\phi(\mathbf{x}_t; t) \right]$$
$$+ \frac{1}{N} \sum_{\mathbf{x}_0 \in \mathcal{D}^-} (1-\alpha) \mathbb{E}_{q(\mathbf{x}_t|\mathbf{x}_0)} \left[ -\log(1 - C_\phi(\mathbf{x}_t; t)) \right], \tag{12}$$

is an importance sampling estimator of the objective function in Eq. (11); proof in Appendix A.5.

**Iterative Training by Stacking Classifiers**   If the classifier is perfect, we know that the DM with score function $s_{\theta,\phi}$ will have improved likelihood and zero infraction (see Corollary 1.1). However, in practice the trained classifier is only an estimate and infractions may not be entirely eliminated.

To alleviate this problem, we note that once the classifier is trained the guided score function $s_{\theta,\phi}(\mathbf{x})$ itself defines a new diffusion model. Consequently, we can employ a similar procedure to train a new classifier on $s_{\theta,\phi}$, aiming to further lower its infraction rate. This iterative approach involves training successive classifiers and incorporating them into the model, progressively enhancing its performance and reducing the infraction rate.

**Distillation**   Adding a stack of classifiers to the model linearly increases its computational cost, since each new classifier requires a forward and backward pass each time the score function is evaluated. To avoid this, we propose to distill the classifiers into the baseline model.

Let $s_{\theta,\mathbf{\Phi}}$ be a "teacher model" consisting of a baseline model $s_\theta$ and a stack of classifiers $\{C_\phi\}_{\phi \in \mathbf{\Phi}}$. We distill $s_{\theta,\mathbf{\Phi}}$ into a new "student model" $s_\psi^{\text{dtl}}$, possibly with the same architecture as the baseline model, by minimizing the following distillation loss

$$\mathcal{L}_\psi^{\text{dtl}} = \mathbb{E}_{\mathbf{x}_0 \sim q(\mathbf{x}_0), t} \left[ \gamma_t \left\| s_{\theta,\mathbf{\Phi}}(\mathbf{x}_t; t) - s_\psi^{\text{dtl}}(\mathbf{x}_t; t) \right\|^2 \right], \tag{13}$$

where $\gamma_t$ is the weight term, similar to the training objective of diffusion models. Here, $\mathcal{L}^{\text{dtl}}$ makes the outputs of the student model match that of the teacher. Algorithm 1 summarises Gen-neG.

## 4   EXPERIMENTS

We demonstrate Gen-neG on three datasets: a 2D checkerboard, collision avoidance in traffic scenario generation for and safety-guarded human motion generation. In each experiment we report a likelihood-based metric on a held out dataset to measure distributional shifts and a kind of infraction metric to measure faithfulness to the oracle.

### 4.1   TOY EXPERIMENT

We first demonstrate Gen-neG on a simple dataset of 2-dimensional points uniformly distributed on a checkerboard grid as shown in Fig. 2. We apply EDM (Karras et al., 2022), a continuous-time DM, to this problem. A baseline DM trained for long enough on this dataset can easily achieve negligible infraction rate. However, because our dataset (see the first panel of Fig. 2) only contains 1000 points the model is prone to over-fitting (see Appendix C.6 for overfitting results). We therefore stop training of the baseline DM before it starts overfitting measured by the evidence lower bound (ELBO) on a held-out validation set. The second panel of Fig. 2 shows samples drawn from the baseline DM and the third panel shows the improved results after one iteration of Gen-neG. We further report in Fig. 3 the rate of oracle violation $\int_{\mathbf{x} \in \Omega^{\complement}} p_{\theta,\phi^*}$, which we will refer to as the infraction rate, and an ELBO estimate by the trained model after each iteration for up to 5 iterations. It demonstrates each iteration

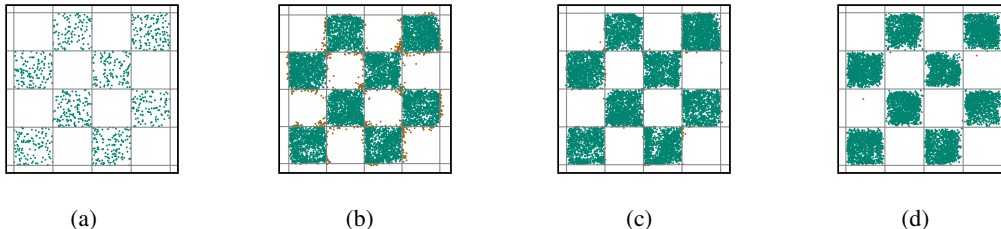

|     |     |     |     |
| :-: | :-: | :-: | :-: |
| (a) | (b) | (c) | (d) |

Figure 2: Samples from the toy experiment. Samples with infraction (i.e. $\mathcal{O}(\mathbf{x}) = 0$) are shown in brown. Fig. 2(a): The true dataset; Fig. 2(b): baseline DM; Fig. 2(c): first iteration of Gen-neG using a label distribution of equal proportions in guidance classifier training; Fig. 2(d): using the wrong distribution to train classifier guidance results in suboptimal density estimation. Here we see that samples are suboptimally pushed inwards from the boundaries. We also have observed that validation ELBOs in these kinds of cases are significantly worse.

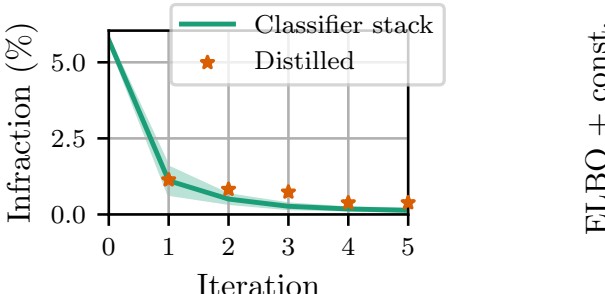
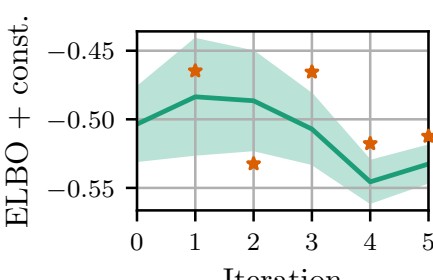

Figure 3: Infraction and ELBO estimations from different iterations of Gen-neG. The infraction rate keeps decreasing, and ELBO remains comparable for the first few iterations. Orange stars show the performance of the model after distillation. We only observe a minor loss of performance in the distilled models. In comparison, without applying importance sampling (the rightmost panel of Fig. 2) reaches an infraction of $0.03\%$ but ELBO of $-1.42$.

of Gen-neG, improves the infraction rate with a comparable ELBO, at least for the first few iterations. Fig. 3 also shows that distillation does not lead to a significant drop in performance.

We also report results for an experiment of one iteration of classifier guidance without the application of proper importance sampling (IS) weights. In this case, the classifier is trained on a synthetic dataset with a uniform class distribution. The last panel of Fig. 2 visualizes samples from this model. While method produces an excellent infraction rate of about $0.03\%$, the classifier severely modifies the shape of the distribution, an undesirable side effect. In particular, we can see the areas close to the boundary have strongly reduced density. In quantitative terms, we find that the ELBO of this imbalanced classifier approach is $-1.42$, a significantly worse result.

### 4.2 INFRACTIONS IN TRAFFIC SCENE GENERATION

We consider the task of traffic scene generation where vehicles of varying sizes are placed on a two-dimensional map with corresponding orientations. Traditionally implemented by as rule based systems (Yang & Koutsopoulos, 1996; Lopez et al., 2018), this task has recently been approached using generative modeling techniques (Tan et al., 2021; Zwartsenberg et al., 2023). In both these bodies of prior work, the common approach has been to discard any samples that violate predefined rules, such as a vehicle being outside the designated driving area ("offroad") or overlapping with another vehicle ("collision"). Rejecting such samples, while effective, can be computationally wasteful, particularly when rule violations occur frequently. Hence, in this context, we employ the Gen-neG to enhance performance. The specific task we consider is to generate $N = 12$ vehicles in a given scene, conditioned on a rendered representation of the drivable area. For each vehicle, the

Table 1: Results for traffic scene generation, in terms of collision, offroad, and overall infractions as well as ELBO. Two varieties ("by-scene" and "by-agent") for the classifier are presented, as well as results with (Gen-neG) and without importance sampling. The final two rows provide the results of distilling the models labelled with † and *.

| Method | Collision (%) ↓ | Offroad (%) ↓ | Infraction (%) ↓ | r-ELBO ($\times 10^2$) ↑ |
|---|---|---|---|---|
| baseline DM | $28.3 \pm 0.70$ | $1.3 \pm 0.14$ | $29.3 \pm 0.64$ | $-27.5 \pm 0.01$ |
| by-scene w/o IS | $20.5 \pm 1.21$ | $0.9 \pm 0.17$ | $21.9 \pm 1.14$ | $-27.7 \pm 0.01$ |
| by-scene | $23.3 \pm 0.7$ | $1.0 \pm 0.28$ | $24.1 \pm 0.67$ | $-27.6 \pm 0.01$ |
| by-agent w/o IS | $14.6 \pm 0.49$ | $0.8 \pm 0.13$ | $15.2 \pm 0.50$ | $-28.0 \pm 0.01$ |
| by-agent† | $16.4 \pm 0.5$ | $0.9 \pm 0.12$ | $17.2 \pm 0.44$ | $-27.7 \pm 0.01$ |
| by-agent stacked* | $11.6 \pm 0.65$ | $0.6 \pm 0.10$ | $12.2 \pm 0.60$ | $-28.0 \pm 0.01$ |
| Distillation of (†) | $12.2 \pm 0.42$ | $0.8 \pm 0.06$ | $12.9 \pm 0.36$ | $-26.8 \pm 0.01$ |
| Distillation of (*) | $5.1 \pm 0.24$ | $0.5 \pm 0.0.09$ | $5.6 \pm 0.20$ | $-27.0 \pm 0.01$ |

position, length, width, orientation and velocity are predicted for a total of 7 dimensions per vehicle. Vehicles are sampled *jointly*, meaning that the overall distribution $p_\theta(\mathbf{x})$ in on $\mathbf{x} \in \mathbb{R}^{N \times 7}$. We train the baseline employing the formalism in DDPM (Ho et al., 2020) with a transformer-based denoising network (Vaswani et al., 2017) on a private dataset. Our architecture consists of self-attention layers and map-conditional cross-attention layers in an alternating order. We use relative positional encodings (RPEs) (Shaw et al., 2018; Wu et al., 2021), which makes use of the vehicles relative positions. Relevant samples (including infracting, and non-infracting ones) and road geometry can be seen in Fig. 1. For our experiments, the oracle function either assigns each scene a collective label, or for each vehicle an individual label. This label is based on the occurrence of collisions or offroad infractions. We then use these results to construct "by-scene" and "by-agent" classifiers.

Table 1 presents the results of our experiments. We provide results for both the "by-agent" and "by-scene" experiments, both showing significant improvement over the baseline result. We moreover show ablations for both settings, where we drop the importance sampling ("w/o IS"), which results in improved infraction rates, but a decreased ELBO. We also provide results for a multi-round approach in the "by-agent" setting, which shows even better results. Finally, we demonstrate that our approach of distilling the resulting models back into a single one works well here too, albeit at a slight drop in performance on ELBO. The presence of lower ELBO can be justified by the classifiers are not trained to optimality and this causes the to deviate from the theory to some extent. On the other hand, training the baseline DM is the only stage where we explicitly maximize the ELBO. Classifiers trained on all the other iterations only implicitly improve ELBO through guiding the model to not allocate probability mass on the invalid region. Overall we find that Gen-neG works as expected, and provides a competitive infraction rate boost over our baseline model, without sacrificing likelihood performance.

## 4.3 Motion Diffusion

Our final experiment is a text-conditional motion generation task on the HumanML3D dataset (Guo et al., 2022). The dataset contains 14,616 human motions annotated by 44,970 textual descriptions. It includes motions between 2 and 10 seconds in length and their total length amounts to 28.59 hours. Each motion is between is between 36 and 196 steps, with the majority of them comprising 196 steps. Each step is represented by a 263-dimensional feature vector, resulting in a dimensionality of over $51,000$ for the largest motions. For our baseline DM, we use the pre-trained checkpoints provided by Human Motion Diffusion (MDM) (Tevet et al., 2023). MDM employs DDPM to learn a transformer-based architecture (Vaswani et al., 2017) with a pre-trained CLIP embedding module (Radford et al., 2021) to facilitate conditioning on the descriptions. It has been shown that, despite its high quality generated motion, under more detailed inspection, MDM and other DMs often lack physical plausibility (Yuan et al., 2022). For example, ground penetration often exists in the generated examples. Such imperfections can cause issues if the model were to be used in downstream applications. To address this issue, we implement an oracle that labels motions with ground penetration at any point in their duration as negative. We then train our classifier with an

Table 2: Results of the Motion Diffusion experiment. "Inf. per step" is the average rate of generated motion frames with infraction while "infraction" is the average rate of generated motions that at least including one infracting frame. r-ELBO is a reweighted ELBO with the same weighting as in diffusion loss.

| Method | Infraction (%) $\downarrow$ | Inf. per step (%) $\downarrow$ | r-ELBO ($\times 10^2$) $\uparrow$ | FID $\downarrow$ |
|---|---|---|---|---|
| MDM | $18.92 \pm 0.58$ | $4.55 \pm 0.03$ | $-6.03 \pm 0.04$ | 0.783 |
| MDM + classifier | $15.35 \pm 0.53$ | $3.15 \pm 0.02$ | $-7.25 \pm 0.11$ | 0.822 |
| MDM + classifier w/o IS | $13.38 \pm 0.50$ | $2.40 \pm 0.02$ | $-9.19 \pm 0.30$ | 1.040 |

architecture that matches that of the original diffusion model with the CLIP encoder removed, because the classifier is not conditioned on the text. We evaluate performance in terms of infraction rate, and the reweighted ELBO, the reweighting referring to a uniform schedule of $\gamma_t$ in Eq. (4).

Table 2 summarizes our results of one iteration of Gen-neG on this dataset. Gen-neG improves both the per-step and overall infraction rate with a small performance drop in terms of the reweighted ELBO and FID. While guiding using a classifier trained without IS weighting produces lower infraction rates, the reweighted ELBO and FID in that case drops even further. Hence, Gen-neG provides a significant improvement of the infraction rates, with a lower cost in terms of model likelihood or sample quality.

## 5   RELATED WORK

Liu et al. (2023) employs diffusion bridges to define a family of diffusion models that are guaranteed to be bound to a constrained set $\Omega$ by construction. Their approach, however, is limited to constraints that admit tractable expectations, rendering it impractical for any but the simplest constraints such as product of intervals in $\mathbb{R}^d$. Kong & Chaudhuri (2023) solves a similar problem dubbed "data redaction." They consider multiple settings, one of which, validity-based approach is the most related to our oracle-assisted guidance, but in the GAN literature. They implicitly perform data redaction by incorporating them into the discriminator and fine-tune the generator. Ansari et al. (2020) utilizes the adept discriminator within the GAN framework, which steers the sampling process through injecting the gradient flow of the $f$-divergence between the real and generated data distributions. However, our particular methodology provides a more comprehensive framework, allowing us to modify the distribution by removing mass from some externally defined negative region. Last, Kim et al. (2022) incorporates a binary classifier in the form of GAN discriminators to refine the learned distribution of DMs, further improving its sample quality. While they utilize a set of tools similar to ours, the problems we tackle are different.

## 6   CONCLUSION

We have proposed a framework to incorporate constraints into diffusion models. These constraints are defined through an oracle function that categorizes samples as either *good* or *bad*. Importantly, such a flexibility allows for simple integration with human feedback. We have demonstrated our model on different modalities demonstrating how it can benefit safety constraints.

The current limitations we recognize, and the possible future directions for this work are (1) incorporating the true training dataset into the later iterations of the method, as the training dataset only affects the baseline DM. The next stages solely use synthetic data. Although we show theoretically that our guidance only improves the model, this lack of revisiting the true dataset in presence of practical errors and approximations poses challenges for large-scale adoption of our method. Our preliminary experiments of visiting the true dataset at the distillation time have not been successful yet. (2) Avoiding stacking of classifiers, instead directly learning an artifact that can replace the previous classifier in our method, similar to (De Bortoli et al., 2021), is vital to the computational complexity of the method as the current computational cost scales linearly with the number of classifiers. (3) Bridging the gap the diffusion bridge-based approaches and our work which is practically applicable to a larger set of applications is another avenue for future developments.

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
