# OpenReview forum: "Don't be so negative! Score-based Generative Modeling with Oracle-assisted Guidance"
_ICLR.cc/2024/Conference — ICLR 2024 Conference Withdrawn Submission_

### Official Review · Reviewer_G3Cq · 2023-10-30

**Soundness:** 2 fair
**Presentation:** 3 good
**Contribution:** 2 fair
**Rating:** 3
**Confidence:** 3

**Summary:**

This paper proposed to suppress negative samples in SGMs. The negative samples are defined by a pre-defined binary oracle. The paper proposed to learn a classifier on positive and negative samples and used the classifier output as the guidance term in SGM. To use multiple classifiers for better invalidity, the paper proposed to distill them into a single model.

The paper evaluated their method on a toy example and two image tasks including a traffic scene generation task and a text-to-motion generation task. Results show that the proposed method achieved better invalidity than baseline diffusion models that did not optimize on this.

**Strengths:**

The major strength of this paper is that the proposed method is simple yet natural and effective. This makes it easy to understand and readers would tend to believe it will wrok. It leverages the advantages of several exsiting techniques: training a classifier from the oracle, using guidance to suppress negative samples, and distillation for better efficiency.

Another strength is the two downstream tasks do face the realistic negative-sample concerns, which further emphasizes the importance of suppressing negative samples.

**Weaknesses:**

One weakness is that there is no theoretical guarantee of the effectiveness of the proposed method. In practice the classifier is not perfect, leading to a bias in the guidance term. It is important to show that infraction can be upper bounded by a small constant if the learned classifier is near optimal. It is also vital to characterize differences for binary classifiers with different Lipschitz constants (i.e. whether the output changes slowerly vs drastically) near the decision boundary. One step further, it is useful to provide a sample complexity bound on the classifier accuracy given number of queries so that we could understand how many samples we need to query the oracle, as it can be expensive in certain tasks.

Another weakness is that there is an obvious tradeoff between infraction and generation quality (especially in the motion experiments), and the infraction rates are not near zero. The last row in table 1 is better than others, but the results can be more persuasive if the numbers are close to zero while the generation quality is similar. The results in table 2 show a considerable degradation of generation quality in compensation for some improvements on infraction.

I'd also point out that some important related works are not discussed in this paper. The *Actively avoiding nonsense* paper by S. Hanneke et al (2018) is the first paper that introduced an active learning algorithm with the help of a validity oracle for suppressing negative samples in generative models. The *Negative data augmentation* paper by A. Sinha et al (2021) is the first paper that applied this idea to deep generative models. The *Safe latent diffusion* paper by P. Schramowski (2022) applied a similar guidance technique to avoid negative samples. The *Erasing concepts from diffusion models* paper by R. Gandikota (2023) first used distillation in this area for enhanced safety. All these works are very related to one or more core ideas proposed in this paper but none of them has been discussed.

There's a typo in table 1 last row.

**Questions:**

Is there any theoretical guarantee of the proposed method mentioned above?

Is it possible to force the infraction to be near zero? If so, what is the generation quality then?

---

### Official Review · Reviewer_DCC6 · 2023-10-31

**Soundness:** 2 fair
**Presentation:** 2 fair
**Contribution:** 2 fair
**Rating:** 3
**Confidence:** 3

**Summary:**

The authors propose a method for generating data which respects some domain-specific conditions. In the paper, the setting in which data is either valid or not is considered. Furthermore, it is assumed to have access to an oracle which can tell, given a data point, whether it is valid or not. The authors propose to train a diffusion model to learn the data distribution, and to use classifier guidance to steer the generation toward the valid region of the data distribution.

**Strengths:**

The paper provides a solution to enforce constraints in the generative process for diffusion models, by reusing standard training of diffusion models and classifier guidance. The concept is relatively simple and straightforward, and the empirical results show effectiveness in reducing the generation of data outside of the support of the data distribution on some simple datasets.

**Weaknesses:**

Some parts of the paper are poorly written and have errors and skipped words. Some sections of the appendix are not referenced in the paper. In my opinion, a reference to works such as [1,2] and a paragraph explaining differences and similarities could be beneficial. The proposed procedure, while improving on reducing the percentage of "invalid" samples, tends to reduce the ELBO, making it an appealing solution only when generating valid samples is crucial even at the expense of the quality of the approximation. The lack of other baselines makes it hard to evaluate the quality of the proposed solution in a broader context.

[1] Liu, Guan-Horng, et al. "Mirror Diffusion Models for Constrained and Watermarked Generation." arXiv preprint arXiv:2310.01236 (2023).

[2] Lou, Aaron, and Stefano Ermon. "Reflected diffusion models." arXiv preprint arXiv:2304.04740 (2023).

**Questions:**

- I don't get the point of going through the classifier guidance procedure. Why wouldn't you directly generate data with the baseline DM and discard the invalid samples by classifying them with the oracle? And if the oracle is too expensive to run at deployment, wouldn't it be more convenient to just distil the oracle with a classifier?
- In the paper, there are no comparisons with other methods. Is it the first time that someone has attempted to solve the problem of generating data that "respect some rules"? Would be interesting to see a more extensive description in the related work and a comparison to other methods such as the referenced "Data Redaction".

---

### Official Review · Reviewer_8oPJ · 2023-11-07

**Soundness:** 3 good
**Presentation:** 3 good
**Contribution:** 3 good
**Rating:** 6
**Confidence:** 3

**Summary:**

This paper suggests a fairly simple method to carefully choose the ratio of “good” and “bad” synthetic samples, determined from some oracle, to be used when training each sequence of guidance classifiers that will guide the generator.

**Strengths:**

The paper has good figures to support the claim and is fairly well structured. The proofs seem to correctly support the theoretical claims made. A wide number of experimental settings were explored making the paper and its proposed method more compelling.

**Weaknesses:**

If the experiments could have more comparison with other discriminator-guided methods that would be helpful to understand the work in a better context. A couple of the proofs in the supplementary seem rather elementary such as the delta method and the relationship between kl divergence and cross entropy. It might be better to just cite textbooks for them.

**Questions:**

One interesting thing the author can do is to explore the relationship between how complex the ideal decision boundaries are and the number of classifiers it would take to guide the generator.

---

### Official Review · Reviewer_6Hw3 · 2023-11-10

**Soundness:** 3 good
**Presentation:** 3 good
**Contribution:** 2 fair
**Rating:** 3
**Confidence:** 3

**Summary:**

The authors propose a method to guide diffusion models to only sample "valid" points by assuming access to an oracle that classifies whether a point belongs to the distribution or not. This setting appears in problems where the produced samples have to respect certain rules (physical laws, security, etc) and it is relatively straightforward to tell if there is some violation. Instead of performing the prohibitively expensive rejection sampling, the authors propose to train classifiers at different noise levels using the generated data from the diffusion model and the annotations from the oracle and then use these classifiers to guide the generation process. The authors validate their method in a self-driving application and in generating human motion.

**Strengths:**

- The problem that the authors study is interesting. In many applications, it is critical to ensure that the generated samples do not violate certain rules.
- The proposed method is intuitive.
- The theoretical results seem correct.
- The presentation of the paper is clear.
- Experimentally, the method seems to achieve its objective in the settings where it is tested.

**Weaknesses:**

- There are no comparisons with baseline methods. It would be nice to include numerical comparisons with other methods that have tried to solve the same problem, e.g. some of the methods mentioned in the Prior Work section.
- The theoretical results are not exactly new. A lot of the proofs use standard results.

**Questions:**

Could the authors include comparisons with baseline methods?